# Organoleptic and Nutritional Traits of Lambs from Spanish Mediterranean Islands Raised under a Traditional Production System

**DOI:** 10.3390/foods11091312

**Published:** 2022-04-30

**Authors:** Rosario Gutiérrez-Peña, Manuel García-Infante, Manuel Delgado-Pertíñez, José Luis Guzmán, Luis Ángel Zarazaga, Susana Simal, Alberto Horcada

**Affiliations:** 1Department of Agronomy, School of Agricultural Engineering, University of Seville, Ctra. Utrera km 1, 41013 Seville, Spain; charo-84@hotmail.com (R.G.-P.); m.garciainfante@gamail.com (M.G.-I.); pertinez@us.es (M.D.-P.); 2Department of Agroforestry Sciences, School of Engineering, University of Huelva, Ctra. Huelva-Palos de la Frontera, s/n, 21819 Huelva, Spain; guzman@uhu.es (J.L.G.); zarazaga@uhu.es (L.Á.Z.); 3Department of Chemistry, University of the Balearic Islands, Ctra. Valldemossa, km. 7.5, 07122 Palma de Mallorca, Spain; susana.simal@uib.es

**Keywords:** light lambs, odour, fatty acids, volatile compounds, autochthonous breed

## Abstract

Our aim was to characterize the organoleptic and nutritional properties of meat from suckling (one-month-old) and light (around three-months-old) lambs in local breeds on the Spanish Mediterranean islands, using meat from male lambs of the Mallorquina and Roja Mallorquina breeds. The lambs were kept with their mother at all times under an extensive management system and fed on mother’s milk until naturally weaned. In the Mallorquina breed, suckling lambs (*n* = 20) were slaughtered after weaning and the light lambs were bred using natural pasture (*n* = 20) or concentrate (*n* = 20), and the Roja Mallorquina light lambs were fed pasture and concentrate (*n* = 20). The pH, colour, texture, water-holding capacity, fatty acids, volatile compounds and sensorial attributes of the meat were analyzed. No differences in meat colour or texture were observed. The highest levels of non-desirable fatty acids were observed in lambs raised using concentrate. Light lambs showed a higher aldehydes content than suckling lambs. High notes of lactic acid and milk flavour were detected. Regardless of access to pasture or concentrate, continued access to mother’s milk during rearing influences the sensorial meat traits of these lambs, so we consider this type of management an optimal way of obtaining the traditional ‘Mediterranean lamb meat’.

## 1. Introduction

Mediterranean autochthonous sheep breeds are adapted to the wide variety of environments found in this type of climate, and there are several commercial categories, depending on the breeds, feeding systems or slaughtering age. The traditional feeding management associated to local husbandry systems has a very strong influence on meat quality [1]. In fact, in order to obtain a homogenous quality of lamb meat and avoid variation in the production, in European Mediterranean areas, lambs are usually slaughtered at a very young age as suckling lambs or light lambs because consumers prefer meat with a low fat content and milder flavour. Meanwhile, in the rest of Europe, lambs are slaughtered at greater weights because the consumers there prefer meat with more intense flavour and a higher fat content. In Mediterranean countries, suckling lambs are slaughtered at around one month old having received only natural mother’s milk or milk substitute as feed (suckling lambs). Light lambs are slaughtered when they are around three months old with a carcass weight of about 13 kg [2]. These lambs are usually separated from their mothers after one month to start on regimes of concentrate, grass and forage. However, according to the traditional system of breeding lambs in the Spanish Mediterranean island breeds, light lambs remain with their mothers throughout rearing while being fed concentrate and/or pasture.

Currently, in Spain, only 10 sheep breeds have high numbers and are not in danger of disappearing. In contrast, a total of 33 sheep breeds are registered as breeds in danger of extinction due to their small numbers. The Mallorquina and Roja Mallorquina breeds are both autochthonous breeds and are labelled as ‘endangered’ by the Official Catalogue of Spanish Livestock Breeds [3]. The populations of Mallorquina and Roja Mallorquina sheep are located on Spanish Mediterranean islands (the Balearic Islands) on 78 and 52 farms, respectively, with average numbers of 150 and 62 adult animals per farm, respectively [3]. Both breeds are rustic, with good mothers and with good dairy conditions. Mallorquina is a breed of lamb bred for meat and Roja Mallorquina is used for the production of both lamb meat and milk, which is used, above all, to produce traditional cheeses on the island of Mallorca. The usual management system is extensive grazing exploitation, making direct use of the natural resources. On some farms, the management can be semi-extensive, and once the lambs are weaned, they are finished off on the same farm.

Traditionally, the feeding systems for sheep from the Mallorquina and Roja Mallorquina breeds are mainly based on the use of natural grazing areas in low-density woodland, with layers of herbaceous plants and shrubs such as *Pistacia lentiscus* or *Erica multiflora* [4]. The management system includes using cultivated pastures, such as oats, ryegrass, barley as green forage (direct grazing or cut grass), dry forage and stubble [4]. In spring and winter, the sheep graze with their young on natural and cultivated pastures, while in summer and early autumn, they take advantage of the stubble and graze on areas of shrubs and trees. To complement the nutritional requirements, mainly during the lactation period and in dry seasons when pasture production is scarce, the ewes receive supplementary feeds including concentrate or grain.

In both the Mallorquina and Roja Mallorquina breeds, the lambs are fed on mother’s milk until 30–45 days (‘suckling lambs’) and after that are raised on pasture with their mothers for two or three months (‘light lambs’), using natural pastureland. Traditionally, in both breeds, the light lambs are slaughtered at around 23 kg live weight because this is expected to produce the maximum meat yield. 

A number of studies on the quality of lamb meat from the main Spanish sheep breeds have been published, including research into the meat quality and nutritional characteristics of the Churra [5], Rasa Aragonesa, Manchega and Merina [2] breeds. All of these breeds have enough animals to be considered foster breeds. However, a significant number of sheep breeds in Spain need to be thoroughly catalogued as they are in danger of extinction. In this context, while there are studies that evaluate the quality of the meat of some autochthonous sheep breeds in danger of extinction such as Grazalema Merino, Churra Lebrijana and Montesina [6], the Mediterranean Island breeds such as Mallorquina and Roja Mallorquina have not yet been studied. In addition, our study was carried out with herds from real farms, where few studies have been conducted into their commercial aspects. On-farm studies contribute to the generation of valuable information from real commercial farms, where not all the factors can be controlled or are often inseparable (e.g., lactation, season and feeding management effects) in contrast to experimental farms or trials. This kind of research is indispensable in order to know how extensive grazing can affect the quality of products [7]. The contribution of local breeds to the maintenance of the rural environment has also been reported: for instance, the traditional system of rearing lambs in the Mediterranean island environment has been described as benefiting the conservation of the Mediterranean natural forest [8]. 

The objective of the current study was to contribute to our knowledge of the meat quality of two autochthonous breeds in danger of extinction linked to the Spanish Mediterranean island environment, with important environmental benefits, such as forest fire prevention, the landscape and biodiversity conservation or carbon sequestration of pastures. This information can be used in plans for the preservation and promotion of the ovine native breeds of the Mediterranean area.

## 2. Materials and Methods

Animal care, handling and experimental procedures were in compliance with international guidelines for European Union procedures on animal experimentation that regulate the protection of animals used for scientific purposes [9]. The lambs were selected according to a similar weight and age in each group and slaughtered using standard commercial procedures, according to the guidelines of Council Regulation on the protection of animals at the time of slaughtering [10].

### 2.1. Animals

An experimental design was proposed, following the traditional production system of autochthonous lambs from the Balearic Islands. We used seventy-four lambs of two autochthonous breeds from Balearic Islands, Mallorquina (*n* = 60) and Roja Mallorquina (*n* = 20) breeds, located around 39°50′25″ N and 2°52′6″ E for the Mallorquina breed, and 39°43′22″ N and 2°56′5″ E for the Roja Mallorquina. According to the traditional system of production on the Balearic Islands, the lambs were not separated from their mothers, so they were able to feed on their mother’s milk until slaughter. Lambs from the Mallorquina breed were raised on three selected farms and were divided into three groups, as follows: ML (*n* = 20), Mallorquina suckling lambs were slaughtered immediately after weaning; MP (*n* = 20), the lambs were raised with their mothers using only natural pasture from the Balearic Islands including natural pastures and oats, ryegrass and barley and cultivated pastures in form of green forage for the three spring months; and MC (*n* = 20), the lambs were raised with their mothers using grain-based cereals (Table 1) and natural pasture from the Balearic Islands including oats, ryegrass and barley in form of green forage for two months. The lambs from the Roja Mallorquina breed (RC; *n* = 20) were raised on two selected farms (10 per farm). Because the birth weight of lambs of the Roja Mallorquina breed is low (<7 kg), lambs are traditionally weaned at the age of approximately two months. After the lambs were weaned, they were raised with their mothers on natural pasture from the Balearic Island, including oats, ryegrass and barley in the form of green forage and grain-based cereal (Table 1) for two months, and then slaughtered at the age of approximately four months. 

### 2.2. Procedures for Slaughter and Muscle Sampling

After refrigeration for 24 h at 4 °C, in the slaughterhouse, the carcasses were weighed and cut into halves along the midline. At the slaughterhouse, the pH values were measured in the *longissimus lumborum* muscle at the 4th–5th lumbar vertebra on the left side of the carcass, using a pH meter (Crison 507; Crison Instruments, Barcelona, Spain) equipped with a penetrating glass electrode. Buffer solutions between 4.01 and 7.0 were used to calibrate the pH meter (Crison Instrument S.A. 08328 Allea, Spain) and temperature compensation was controlled during the test. 

After the carcasses were chilled, the left halves of the carcasses were taken to the laboratory where the longissimus *thoracis-lumborum* (LTL) muscle was extracted and sliced. The proximate composition of LTL muscle was measured according to standard AOAC procedures [11] and expressed on a wet basis: the moisture content by procedure 24003, the nitrogen content by procedure 2057, the intramuscular fat content by procedure 13032, and the ash content by procedure 14066.

At 48 h post-mortem, the colour of the LTL was evaluated after one hour of blooming using a Konica-Minolta CM colorimeter (Konica Minolta Inc., Tokyo, Japan) in the CIELAB color space [12] with a standard D65 illuminant, observed angle of 10°, and zero and white calibration. The average of the three different measurements taken on the same surface (between lumbar and thoracic regions) were obtained. The lightness (L*), redness (a*) and yellowness (b*) were recorded. The Hue angle (H°) was calculated as tan^−1^(b*/a*) and the chroma (C*) as [(a*^2^ + b*^2^)^1/2^]. Two portions of LTL muscle (5 g each) were used to determine the water-holding capacity (WHC) of the meat according to Guzmán et al. [13] using the pressure method. The WHC was expressed as the percentage of juice expelled after compression. A portion of 10 g from the LTL muscle was used to calculate the haem pigment concentration using a physical–chemical method expressed as mg myoglobin/g fresh meat [14]. The rest of the LTL muscle was vacuum-packed into three packages and frozen at −20 °C before measuring the Warner–Brätzler shear force (WBSF), fatty acid profile and volatile compounds.

To determine the WBSF, a vacuum-packed sample of LTL was thawed overnight at 4 °C. The sample, including a portion of caudal slices of LTL, was heated in a 75 °C water bath to an internal temperature of 70 °C. The temperature was monitored with a Jenway thermocouple equipped with a probe (Hanna Instruments HI 8757; 20600 Eibar, Gipuzkoa, Spain), and a Stevens QTS 25 texture analyser equipped with a WB device [13] was used. The LTL muscle was cut into slices with a cross-section of 1 cm^2^ parallel to the muscle fibres. The maximum shear force (kg/cm^2^) was assessed in at least three subsamples of heated meat pieces parallel to the muscle fibres. Analyses were run in duplicate.

### 2.3. Fatty Acid Analysis

Intramuscular fatty acid methyl esters (FAMEs) were analysed following the method proposed by Gutiérrez-Peña et al. [15]. A sample of approximately 1 g of LTL was thawed and saponified in 6mL of 5M KOH in methanol:water (50:50 *v*/*v*) with hydroxyquinone (1 g L^−1^) at 60 °C for 1 h, after flushing with nitrogen, after which the mixture was diluted with 12 mL of 0.5% NaCl and 5 mL of petroleum ether and the non-saponifiable fraction was removed. To neutralise the KOH, 3 mL of glacial acetic acid was added. Double petroleum ether clearance was used to extract the FAMEs, with nitrogen used to evaporate the solvent. The extracted FAMEs were then methylated using 200 µL of TMS-DM in methanol:toluene (2:1, *v*/*v*) at 40 °C for 10 min, dried under nitrogen and dissolved in 1 mL of n-hexane containing 50 ppm of butylated hydroxy toluene. The samples were centrifuged at 15,000 rpm for 5 min and the supernatant was transferred for analysis. The separation of FAMEs was carried out using a gas chromatograph Agilent 6890N Network GS System (Agilent, Inc., Santa Clara, CA, USA) equipped with a flame ionisation detector (FID) and fitted with a HP-88 capillary column (100 m, 0.25 mm i.d., 0.2 μm film thickness, Agilent Technologies Spain, S.L., Madrid, Spain). The samples were automatically injected using an HP 7683 injector. The chromatographic conditions were as follows: initial column temperature 100 °C, programmed to increase at a rate of 30 °C/min to 158 °C and then at 1.5 °C/min to 190 °C, maintaining this temperature for 15 min, then at 2 °C/min to 200 °C and then increasing again at 10 °C/min to a final temperature of 240 °C for 10 min. The injector and detector were kept at 300 and 320 °C, respectively. Hydrogen was used as the carrier gas at a flow rate of 2.7 mL/min. The split ratio was 17.7:1, and 1 µL of solution was injected. Nonadecanoic acid methyl ester (C19:0 ME) at 10 mg/mL was used as an internal standard. Individual FAMEs were identified by comparing their retention times with those of authenticated standards from Sigma (Sigma Chemical Co., Ltd., Poole, UK). The fatty acid profile of the intramuscular fat was expressed as a percentage of the total fatty acid detected.

### 2.4. Volatile Compound Analysis

The profile of meat volatile compounds was studied using the solid-phase microextraction (SPME) analysis technique. A sample (20 g) of the LTL muscle was thawed at 4 °C overnight before analysis and was cooked at 200 °C under a mixed closed griddle (Jatta electro, GR266 1000W, Abadiano, Vizcaya, Spain). The grill was switched on for 15 min before the sample was grilled. Each sample was placed in the middle of the grilling tray in order to be grilled uniformly and was cooked for approximately three minutes at a temperature of 70 °C [16]. Directly after cooking, the meat was chopped finely and, together with all the fat released from the steak during cooking, placed in an electric bowl chopper (Janke and Kunkel A-10, IKA Labortechnik, Germany). Straight after chopping, 10 g of the cooked sample was placed in a headspace vial (Tekmar, 100 mL) and equilibrated for 40 min at 40 °C prior to exposure of the SPME fibre (Fibre Assembly 50/30 μm DVB/CAR/PDMS, Stableflex-2cm-23Ga, Gray-Notched; Bellefonte, Pensilvania, EEUU) placed over the sample for a further 20 min. The analysis of volatile compounds was performed using a Thermo Scientific TRACE 1300 series (Milan, Italy) gas chromatograph (GC) equipped with a Thermo Scientific TRIPLUS RSH autosampler (Milano, Italy) for injection and coupled with an ion trap mass spectrometer (Thermo Scientific ISQ QD Single Quadrupole Mass Spectrometer; Milan, Italy). The desorption process of the compounds was carried out using the spitless mode with purges of 5 mL/m. Injection was maintained for 10 min at a temperature of 40 °C. Meanwhile, the chromatograph oven was heated to 40 °C and, once the sample had been released into the gas phase, the temperature was increased at a rate of 5 °C/min up to a maximum temperature of 220 °C. Helium at 20 psi with a flow of 1.2 mL/min at 40 °C as carrier gas was used. The volatile compounds were separated using a VF-WAXms fused silica capillary column (30 m length × 0.25 mm id × 0.50 μm film thickness, Agilent Technologies, Inc. 2012, Santa Clara, CA, USA). The operating conditions were as follows: initial temperature, 40 °C for 5 min, then increased to 220 °C at a rate of 5 °C/min and held for 5 min, with a total acquisition time of 56 min. The temperatures of the source and quadrupole were 175 °C and 150 °C, respectively. The mass spectra of volatile compounds were generated by a MS equipped with an ion trap. The data acquisition was performed by scanning the mass range 29–400 amu. in EI mode (70 eV with emission current of 50 mA) at 1.9 microscan/s. The volatile compounds were identified by comparing their mass spectra with spectra included in NIST/EPA/NIH Mass Spectral Libraries and confirmed by matching their LRI with Resconi et al. [17,18] and Elmore et al. [19] or the Flavornet database. The linear retention indices (LRI) [20] were calculated by previous injection of standards of saturated n-alkanes (C6-C22) under the same GC–MS conditions. The volatile compounds were expressed as a percentage of the total volatile compounds identified.

### 2.5. Sensorial Evaluation

For the sensorial evaluation, a 10-member trained panel was used. In order to recognize the sensorial parameters in meat lambs, special training was undertaken before beginning the process of evaluating the samples, in which the panel chose from a chart of 21 sensorial features those most related to the samples to be assessed. The parameters selected were: the intensity of the lamb, milk and liver odours, toughness of the meat, initial and final juiciness, friability, chewiness, and the intensity of the lamb, lactic acid and liver flavours. The *semitendinosus* muscle was separated from the lamb’s leg and packed in aluminium foil with three of the four sides of the package closed. A penetration K-type thermocouple (HI-766Z/7, Hanna, 20600 Eibar, Gipuzkoa, Spain) connected to a thermometer (HI-93552R, Hanna, UK) was placed in the centre of the muscle, which was then cooked until a temperature of 70 °C was reached in the centre by placing it on a two-sided grill (Asteria 2200 W, Taurus, Barcelona, Spain) previously pre-heated to 250 °C. Then, discarding the ends of the muscle, the muscle was divided into 2 cm × 2 cm × 1.5 cm pieces and immediately served to a trained panel [21] in Petri dishes of 55 mm diameter. The trained panellists were asked questions about the intensities of the lamb, milk and liver odours; hardness, initial and final juiciness, friability, chewiness; intensities of the lamb, lactic acid and liver flavours; final juiciness and persistence of the global flavour. Sample presentations include water, apple and unsalted bread to neutralize the flavours in the mouth. For each attribute, the description and procedure were provided, together with a 10-point scale for the evaluation, where 0 indicates that the sample does not present the considered attribute, 1 that it is presented with a very low intensity, and 10 that it is presented with a very high intensity. The results were expressed as the median and standard deviation.

### 2.6. Statistical Analysis

A statistical descriptive analysis was carried out to describe the meat-quality parameters of the Mallorquina and Roja Mallorquina breeds reared under a traditional production system. The effects on meat quality of the feeding system in each breed studied were analysed using the GLM procedure and a post hoc Duncan test, considering probability values below 5% as significant in comparison with the means test. The carcass weight was used as a linear covariate. In order to assess the differences among the ML, MP, MC or RC groups of lambs, and to determine the contribution of meat quality variables to these differences, a canonical discriminant analysis including all the data variables was performed. Finally, to identify the variables that contributed to a greater extent to the differentiation of the groups of lambs, a stepwise regression discriminant analysis was carried out. The Wilks’ lambda method shows which variables contributed significantly to the discriminant function. The variables were selected when a variable explains a part of the observed variability (*p* < 0.05). Selected variables were included in a multinomial logistic regression to calculate the probability of a lamb carcass belonging to ML, MP, MC or RC. All the data were analysed using StatSoft, Inc. (2014), STATISTICA software (data analysis software system), version 12 (www.statsoft.com, accessed on 15 May 2021, Tulsa, OK, USA)

## 3. Results and Discussion

### 3.1. Carcass Weight and pH Value of the Meat

Carcass weights from suckling lambs of the Mallorquina breed (Table 2) are included in the commercial category of Spanish lamb breeds [22]. In fact, suckling lambs of the Mallorquina breed (ML) presented a 6.7 kg average carcass weight, as reported by Campo et al. [2] for the Manchega, Merina or Rasa Aragonesa breeds. However, the carcass weights from light lambs were lower than that reported for the Spanish market by Campo et al. [2] in the light lamb category (range 11.5 to 12.91kg). This observation is particularly important in the lambs of the Mallorquina breed fed in their traditional system on Mediterranean island pastures. Here, the carcass weight of the lambs raised on pasture was approximately 24% lower than that of the lambs raised on concentrate feed. These differences between the two management systems have been found in other studies with autochthonous Mediterranean breeds [23]. Differences have also been reported in the characteristics of sheep meat between northern and Mediterranean European countries [22]. These differences are due to the fact that sheep’s meat from the Mediterranean region is derived from young animals with a mild flavour in contrast to that from the northern regions, which is derived from older animals and has a more intense flavour. 

WHC: water holding capacity, expressed in percentage of liquid expelled. The pH values measured in meat from the Mallorquina lamb breed were in the range of 5.64 to 5.89. These values were similar to the range of values found by Madruga et al. [24] after 24 h for the lamb meat (range 5.97 and 6.04), indicating that transporting the animals to the slaughterhouse and their handling prior to slaughter were correctly carried out for rigor mortis to set in and transform the muscle into meat. Although the lambs were not slaughtered on the same day, the pH values observed in the meat were not affected by the slaughter routine of the lambs, since conditions were taken into account in the slaughterhouse about the protection of animals at the time of slaughtering. However, it should be noted that significant higher pH values were observed in meat from the Roja Mallorquina breed (6.08) than that of the Mallorquina breed (*p* < 0.05). The Roja Mallorquina breed is probably more susceptible to stress factors at the slaughterhouse than the Mallorquina breed, leading to a late increase in the pH of the meat [25].

### 3.2. Proximate Composition

No significant differences among meat from all the feeding systems in moisture, ash and fat content were found. These results agree with Campo et al. [2] that reported moisture and ash contents are usually highly homogeneous in lamb and goat meat. The absence of significant differences among the fat content observed among suckling and light lambs is probably due to the low carcass weight at which light lambs are traditionally slaughtered in the Balearic Islands. However, significant differences in the protein content in meat were observed (*p* < 0.05). In fact, a lower protein content in meat from the Roja Mallorquina breed was observed than in the Mallorquina breed under a similar feeding system based on the use of concentrate (Table 2). In the Spanish market, the differences in cold carcass weight between the commercial categories of lambs are a reflection of their slaughter age (30 and 70 days for suckling and light lambs, respectively) and the system of feeding (suckling for animals that are only fed milk or light lambs that are fed only on solid feed and forage) [22]. However, according to traditional feeding system of production in the Spanish Mediterranean islands, light lambs are fed using solid food and mother’s milk. In this context, a similar composition of the meat from suckling and light lambs from Mediterranean breeds could be expected.

### 3.3. Technological Meat Quality Traits

The colour values of Mallorquina and Roja Mallorquina meat refrigerated in aerobic storage for 48 h are shown in Table 2. The production system used in both breeds did not affect the meat colour (*p* > 0.05). Although the myoglobin contents observed in the meat of the ML and RC groups (3.68 and 3.04 mg/g fresh meat, respectively) were lower than that observed in the light lambs fed on concentrate or grass, these differences were not significant (*p* > 0.05). Several papers have reported a high content of myoglobin in the meat of heavier lambs raised on pasture vs. suckling lambs, due to a higher content of Fe through the synthesis of myoglobin molecules obtained mainly from grass. However, in traditional rearing systems for Mallorquina lambs, suckling lambs and those fed with concentrate are kept together with their mother using mainly areas of pasture. Additionally, a wide variability in myoglobin values for each treatment has been observed. It is important to note here that the wide variability within each group indicates the reduced homogeneity of meat colour within each animal group and could affect comparative statistical results. The results observed for the trichromatic values of the meat in light lambs in the Mallorquina and Roja Mallorquina breeds were within the range reported by Mateo et al. [5] for suckling Churra lambs and Martinez-Cerezo et al. [26] for meat from light Rasa Aragonesa, Churra and Spanish Merino breeds. The lack of colour difference found in this study can be explained by a lack of differences in the Fe or haematin content among suckling and light lambs. Values between 34 and 42 have been suggested for L* in aerobically stored lamb meat as a determinant for consumer acceptance for Australian meat [27] and for the meat of 27 kg Spanish fattened lambs [28]. It therefore follows that we can assume that Mallorquina and Roja Mallorquina meat lies within an acceptable range of L* to be accepted in the Mediterranean market. No differences in lightness, redness, yellowness, chroma and hue were observed in meat among suckling lambs, grazing and concentrate feeding lamb groups (*p* > 0.05).

In the Spanish lamb market, consumers show a marked preference for a meat colour classed as pale, which is obtained from light lambs [29]. All the feeding systems could change the colour of the meat; however, the lamb production system in the Balearic Islands based on the use of pasture does not produce changes in meat colour, for two main reasons. Firstly, the animals present a lower Fe content in the meat, since they remain with their mothers feeding on breast milk with supplements such as concentrated food or grass, and secondly, they are slaughtered at an early age.

The shear force values of the meat from all the feeding systems agreed with those reported by Mateo et al. [5] in suckling lambs or Carrasco et al. [30] in light lambs from the Churra breed and correspond to the preference for light lamb in the Mediterranean market. However, the WBSF of the cooked lamb meat was different among treatments (*p* < 0.05; Table 2). Higher values of WBSF were observed in meat from MP lambs than suckling lambs or lambs raised using concentrate. This observation may be related to a greater movement of MP lambs in the grazing areas to ingest the grass compared to suckling lambs or light lambs that received concentrated feed from the feeder. The fact that the meat obtained from grazing animals turned out to be tougher and firmer could be explained by a greater muscular development and higher connective tissue content of the LTL muscle, since the animals were more active as they obtained food during grazing [31,32].

Cooking losses were not affected in any of the feeding systems (Table 2; *p* > 0.05). Carrasco et al. [30], comparing different lamb diets, found no differences in cooking losses, whereas Santos-Silva et al. [33] reported higher cooking losses on forage diets vs. concentrate diets. As was reported by Kemp et al. [34], the cooking losses were mostly due to differences in fat content. In fact, Carrasco et al. [30] indicated that fattier meat would have less evaporative losses while leaner meat would have more cooking losses. In the meat in the current study, there were no differences in the intramuscular fat content among the different meats from the Mallorquina and Roja Mallorquina breeds under different diets, which agreed with the lack of differences in cooking water losses observed. However, the influence of the different diets on pressure losses was observed (Table 2; *p* < 0.001). In fact, the concentrate diet showed, in both Mallorquina and Roja Mallorquina breeds, higher water losses than the meat from suckling and pasture-fed lambs. Unlike cooking losses, pressure losses are important in fresh meat. The state of water in fresh meat is related to the OH^−^ and H^+^ bonds in water with the protein fraction of the muscle. The higher water losses (lower WHC) in the meat of the lambs raised on concentrate (in both breeds studied) may be related to the lower relative protein content observed in meat from MC and RC lambs vs. meat from suckling lambs or lambs raised on grazing.

### 3.4. Fatty Acid Composition

Descriptive statistics of the main fatty acids of intramuscular fat in the Mallorquina and Roja Mallorquina breeds are presented in Table 3. The fatty acid profile was within the range of those reported by Juárez et al. [6] for Grazalema Merino, Churra Lebrijana, Merino, Montesina and Segureña Spanish breeds and Sañudo et al. [35] for the Rasa Aragonesa breed, with C18:1 being the most abundant fatty acid in all cases.

Significant differences in SFA content among different systems used on the Balearic Islands were observed (*p* < 0.001). Although SFA are associated with milk intake, a higher SFA content in intramuscular fat was observed in light lambs in the Mallorquina breed (range of 49–50% of the total fatty acid detected) that were raised on pasture or concentrate than suckling lambs. The most favourable range of SFA for human health in meat was observed in ML and RC lambs (*p* < 0.05) coinciding with that described by Urrutia et al. [36] in autochthonous Navarra breed lambs. Palmitic acid (C16:0) was the most abundant SFA from the total SFA. C16:0 is considered a non-desirable fatty acid and is associated with a concentrate diet [37]. However, a lower C16:0 content was observed in suckling lambs from the Mallorquina breed and Roja Mallorquina light lamb breed raised using concentrate.

The MUFA content in the meat (Table 3) was in line with that proposed by Arana et al. [38] in the Rasa Aragonesa breed. The lowest total MUFA and mainly C18:1 fatty acid content was observed in the Mallorquina breed raised on pasture, while a higher MUFA was observed in suckling lambs and light lambs raised using concentrate (MC and RC). The MUFA content has been reported to be associated with using grain or concentrate diets [39]. This is in line with the literature, which reports that MUFA are preferentially incorporated into the triglycerides of the body’s fat depots in animals fed on concentrate [40].

The PUFA content in the meat of lambs of the Mallorquina and Roja Mallorquina breeds was higher than that shown by Campo et al. [2] in different lamb cuts of the Rasa Aragonesa breed. These authors described a content ranging from 6.21% of PUFA content in the breast portion to 7.95% of PUFA in the leg portion of light lambs, while ranges between 16.04 and 19.75% of PUFA were observed in meat from the lambs of Mediterranean breeds raised under a traditional feeding system based on the use of mother’s milk and pasture or concentrate. A higher PUFA content was observed in meat from light grass-fed lambs and suckling lambs than in meat from concentrate-fed lambs (Table 3). The literature consulted refers to the higher content of PUFA in meat produced in feeding systems in which grazing is the main feed resource. Specifically, the type and quality of pasture affects the ratio of 18:3 fatty acids in meat because linolenic acid (18:3 n-3) has been shown to be one of the main components of the total lipid content of grass [41]. In our study, the higher C18:3 n-3 in MP and ML than lambs raised using concentrate was observed *p* < 0.05; Table 3). On the other hand, as reported by Wood and Enser [42], grass lipids are found inside organelles (chloroplast) that remain intact during the ruminal digestion processes, thus providing a natural protection for n-3 PUFAs against biohydrogenation in the rumen. The high content of PUFA observed in the meat of suckling lambs may be due to the fact that in the traditional production systems of the Spanish islands of the Mediterranean area, the main source of food for ewes is grass, and consequently, the PUFA content in mothers’ milk is expected to be high. 

The consumption of lamb and other red meat is associated with diets with a high fat content, especially those containing saturated fatty acids. Indeed, certain saturated fatty acids have been linked to cardiovascular disease [43] while an intake of PUFA (mainly n-3 PUFA) is recommended to prevent coronary artery disease. Recommendations about heathy eating include reducing n-6/n-3 ratios in the human diet. In fact, public health recommendations advise consuming a maximum of five n-6 PUFA to every one n-3 PUFA [44]. The n-6/n-3 nutritional quality indicator related to the health of consumers in the meat of lambs from the Spanish islands of the Mediterranean raised in the traditional way has shown favourable values of less than 5. According to the recommendations proposed by Salter [43], only the production system of the Mallorquina breed, which is based on the use of concentrated feed, has shown an unfavourable ratio for the consumption of lamb meat (slightly higher than seven n-6 PUFA to one n-3 PUFA). The Atherogenic index (AI) can be considered a suitable measure of the nutritionally healthy value of meat [45]. Usually, ranges of 0.5 to 1 in meat fats are reported [46], while values of <0.5 have been described in vegetable oils. In meat from Mallorquina and Roja Mallorquina lamb breeds, the AI (0.72–0.93) was higher than the range described by De Sousa et al. [47] in the meat of Churra lambs (0.71) or suckling kids (range 0.70–0.75) of four autochthonous Spanish breeds. Significant differences in the AI between light lambs were also observed. According to Sinanoglou et al. [48], differences in AI are mainly associated with genetic and feeding factors. The best AI was observed in the meat of the light Roja Mallorquina lamb breed and suckling lambs from the Mallorquina breed.

### 3.5. Volatile Compounds

A total of 66 volatile compounds with different chemical compositions in meat from the Mallorquina and Roja Mallorquina breeds were identified. Table 4 shows the most abundant volatiles identified, classified into nine chemical families. 

Most of the compounds detected in cooked meat from the Mallorquina and Roja Mallorquina breeds come from the lipid oxidation reactions (aliphatic aldehydes, ketones, hydrocarbons, and alcohols) and Maillard reaction (pyrazines, pyrroles, carboxylic acids, furans and other sulphur compounds) [50]. Most of the volatile compounds obtained in the lamb meat in the current study were included in aliphatic aldehydes (*n* = 11 volatile compounds and total relative percentage abundance in the range of 67.48–78.33%), aliphatic ketones (*n* = 13 volatile compounds and relative percentage abundance in the range of 5.93–8.53%) and aliphatic alcohols (*n* = 9 volatile compounds and total relative percentage abundance in the range of 5.25 to 7.37). These ranges are in accordance with the relative percentage abundance showed by Insausti et al. [51] (84.79 of relative percentage abundance) in light lambs from the Rasa Navarra breed raised on concentrate and forage. The most frequently represented compound in the lamb meat from the Mallorquina and Roja Mallorquina breeds is hexanal, as has been observed in lambs of other native breeds such as Rasa Navarra [51] and Churra [52]. According to the Flavornet database, this aliphatic compound is said to provide aromas related to green grass, and a fruity smell in meat [53]. The relative content of octanal, nonanal and 3-hydroxy-2-butanone has been high in the meat of all the animals studied. While these compounds are identified in the meat of all the groups of lambs studied, significantly higher values have been observed in the meat of suckling lambs [52]. Some furans are reported to be formed by the oxidation of PUFA [54]. In fact, furfural is postulated to be a flavour compound derived from the oxidation of C18:3n–3, which lends herbal notes to meat [54]. A significantly high relative percentage abundance of furfural in meat from Roja Mallorquina breed (*p* < 0.05) was observed.

As regards the chemical families produced by the Maillard reaction, sulphur compounds accounted for between 2.26 and 5.56% of the total relative percentage abundance, including three volatile compounds identified. Sulphur compounds have been related to the characteristic flavour of lamb meat and are identified at low detection levels. The highest total relative percentage abundance of sulphur compounds was observed in meat from suckling lambs from the Mallorquina and light Roja Mallorquina breeds raised on grain. These compounds have low detection thresholds, and their distinctive odour characteristics produce sweet notes in sheep meat [17].

Finally, the highest contents of chemical families of pyrazines, pyrroles and carboxylic acids were observed in meat from the Roja Mallorquina breed. These families represented 21 out of the total 66 compounds identified. Among these compounds, 2-ethyl-3,(5 or 6)-dimethylpyrazinea and 2,3-diethyl-5-methylpyrazinea are of particular importance as they are said to provide notes of shrub, grilled meat, greenery, toast and earth flavours [17].

### 3.6. Sensorial Analysis

Table 5 shows the sensory results evaluated by the trained panel. In general, the scores ranged between 1.28 and 8.1 on a scale of 10 points in terms of odour, flavour and texture meat traits. The highest scores were observed for the intensity of texture traits (chewiness, friability and juiciness) and the milk odour of meat, while the lowest scores were observed in the sensory values in relation to the intensity of other odours and hardness in meat. These results may be in agreement with the type of lamb meat usually marketed in Spain, where the animals are slaughtered at an early age to avoid a strong odour and flavour that is not desired by consumers [30,55]. The main reason for rejecting lamb meat by consumers in the Mediterranean area of Europe is probably the strong odour and taste of the meat. Here, the lamb meat of the Mallorquina and Roja Mallorquina breeds has been found to have medium values (around 6 on a 10-point scale) as the smooth intensity of the odour and flavour to lamb or lactic acid, while the reduced intensity of flavour to liver stands out (range 1.28 to 2.10 on a 10-point scale). In all groups, the milk odour score (range 6.34 to 6.79) was higher than the characteristic lamb odour score (range 5.96 to 6.19). This observation may be related to the idea that the traditional feeding system for lambs of the Mallorquina and Roja Mallorquina breeds includes access to breast milk throughout the time the animals are reared. In fact, meat from suckling lambs has shown significantly higher values of milk odour (6.97) intensity than that observed in light grass-fed (6.34) or concentrate-fed lambs (6.52 and 6.76 to MC and RC, respectively). 

The highest lamb flavour was observed in MP lambs (6.94 on a 10-point scale). Aromatic compounds from the grass could persist in the meat of lambs raised on pasture. Some volatile compounds, such as 2, 4-decadienal, develop from the unsaturated fatty acids such as α-linolenic acid present in grass [19]. According to Resconi et al. [56], this volatile compound has a reduced level of sensibility and can easily be perceived sensorially. The intake of α-linolenic acid and other unsaturated fatty acids by light lambs feeding on grass could account for the greater lamb flavour observed in meat from Mallorquina lambs raised on pasture on the Spanish Mediterranean Islands.

Sensory analyses on the Mallorquina and Roja Mallorquina breeds raised in traditional system productions showed that lamb flavour is affected fundamentally by the use of grass when the lambs are feeding. Even though all the light lambs received mother’s milk, significantly higher sensorial notes were observed in the MP group. Sensory analyses in the Rasa Aragonesa breed have shown that the lamb flavour is affected by the slaughter weight and feeding system [55]. However, the lack of differences in lamb odour or milk odour between the Mallorquina and Roja Mallorquina breeds among the group studied was probably due to the animals being young and lean and the fact that all the animals received mother’s milk as part of the traditional production system.

Regarding the texture properties, consumers of lamb meat expect a tender, juicy meat that feels soft on the palate. The hardness attribute in Mallorquina and Roja Mallorquina meat showed low levels (between 3.44 and 3.78). Although the animals had ample freedom of movement together with the mothers, the meat can be considered tender since the lambs were slaughtered at an early age and there was little development of the muscle tissue. Meat hardness in animal species is generally greater in older animals [57]. According to Sañudo et al. [58], the variation in meat hardness values in terms of the slaughter weight depends mainly on the myofibrillar content rather than the total collagen content or its solubility, especially when the toughness of cooked meat is considered.

The range of values for the final juiciness of the meat (range 6.82–7.31) was higher than those for initial juiciness (range 5.94–6.31). When the meat first enters the mouth, the juiciness evaluated (initial juiciness) was similar for all the groups of meat analysed, while the final juiciness scores were higher in the light lambs than in the suckling lambs. The in-creased levels of fat were also associated to significant increases in the final juiciness of the cooked meat. 

The best sensory traits observed in the Mallorquina and Roja Mallorquina breeds were linked to the friability and chewiness of meat, because scores above 7.2 points out of 10 were recorded. Significant differences in the chewiness of meat in different groups of lambs were not observed. The lowest values for friability were observed in meat from lambs raised using milk and grain (*p* < 0.05). In general, regardless of the traditional production system for lamb meat from the Mallorquina and Roja Mallorquina breeds, the judges considered that the meat has high friability and requires little chewing to be swallowed.

### 3.7. Discriminant Analysis

The results of the discriminant analysis including all the data variables are shown in Figure 1. Each animal group is clearly placed in each quadrant of the plot. Root 1 separates light Mallorquina lambs raised on pasture or concentrate from suckling lambs or the Roja Mallorquina breed. Root 2 separates ML and MP from MC and RC groups. ML and MP are placed close together because both groups had high availability of access to mother’s milk. Normally, grazing lambs remain with their mothers during the day, taking advantage of the pastures, and are rounded up at night in the sheepfold with their mothers, where they continue to feed on their mother’s milk. The use of concentrated feed in the rearing of lambs allows us to situate the MC and RC groups in the lower part of the plot.

Table 6 shows the best selection of variables chosen to discriminate meat from the Mallorquina and Roja Mallorquina breeds raised on their traditional production system in the Spanish Mediterranean Islands. Seventeen variables were included in a multinomial logistic regression to calculate the probability of a carcass belonging to their group of traditional production. According to the matrix classification proposed including these seventeen variables, 90.21 and 82.43% of the carcasses were correctly classified into ML and MP, respectively, while 100% of carcasses were correctly classified into MC or RC. The best variables selected (higher F value) to predict belonging to a group of lambs raised are linked to the polyunsaturated fatty acid (*p* < 0.001) and monounsaturated acid content (C18:1n-9c and 17:1; *p* ≤ 0.001). According to Enser et al. [59], a high content of polyunsaturated fatty acids has been described in meat from animals raised on pasture, while a high content of monounsaturated fatty acids is observed in the meat of lambs fed using concentrates. 

The parameters related to the juiciness of the meat (WHC and final juiciness) are the variables that best mark the discrimination of the different traditional systems for raising lambs from the Spanish Mediterranean. Among all the aromatic compounds detected in the meat of the reared lambs, aromatic hydrocarbons are the ones that best mark the differentiation of the traditional systems of lamb meat production in the Balearic Islands. Finally, saturated fatty acids such as C10:0 and C12:0 contribute to the differentiation of the production systems of lambs that have fed on mother’s milk, while high contents of saturated short- and medium-chain fatty acids have been described in ruminant milk [60].

Traditionally, the characterization of livestock breeds has been studied from the genetic point of view and from the assessment of the quality of their products. This has been the way to recognize the origin of livestock breeds and value the quality of the products of local breeds. However, new technologies based on knowledge of muscle protein are now being used to monitor systems of animal production and their product quality. In fact, proteomic investigations [61] have been reported in improved livestock breeds with the aim of gathering relevant information to help improve the meat quality in cost-effective and efficient production systems or even to monitor specific differences among breeds [62]. Advanced proteomic techniques to help characterize production models and identify the origin of animal products are usually applied to improved livestock breeds. However, in the future, these techniques could be used in local breeds with a reduced number of flocks, as has occurred in the breeds of Spanish Mediterranean islands.

## 4. Conclusions

This is the first time that the organoleptic characteristics and nutritional values of lamb meat from native breeds in danger of extinction (Mallorquina and Roja Mallorquina) reared in the traditional system on the Spanish Mediterranean islands have been described. Regardless of access to pasture or concentrate, continued access to mother’s milk during rearing influences the sensorial meat traits of lambs from the Spanish Mediterranean islands. The pH of the meat was not affected by the traditional system of raising lambs for the Spanish Mediterranean island breeds. The meat colour was the light-pink-coloured meat appreciated by Southern European consumers, while there was no evident effect on meat colour of including pasture or grain in the feeding of light lambs. The same tendency was observed in meat texture: there was no evident hardness, as the lambs are slaughtered early. The fatty acid composition values from the lambs were comparable, in quality and nutritive values, to other commercial autochthonous Spanish breeds. The low intramuscular fat content, low atherogenic index, and high PUFA content suggest that meat from Mediterranean lamb breeds is highly suited to human consumption. The aliphatic compounds (hexanal, nonanal, octanal and 3-hydroxy-2-butanone) detected in meat from Spanish Mediterranean island breeds are seen to produce aromas related to sensorial notes of green grass, milk and fruit. 

Meat from Mallorquina and Roja Mallorquina lambs raised in their traditional production system conforms to the standard lamb meat acceptability of southern European consumers. In the future, these Spanish Mediterranean island breeds could be monitored by proteomic methods and a Geographical Protected Indication label could be developed under the EU Program to guarantee the traditional rearing practices and origin of the breed.

## Figures and Tables

**Figure 1 foods-11-01312-f001:**
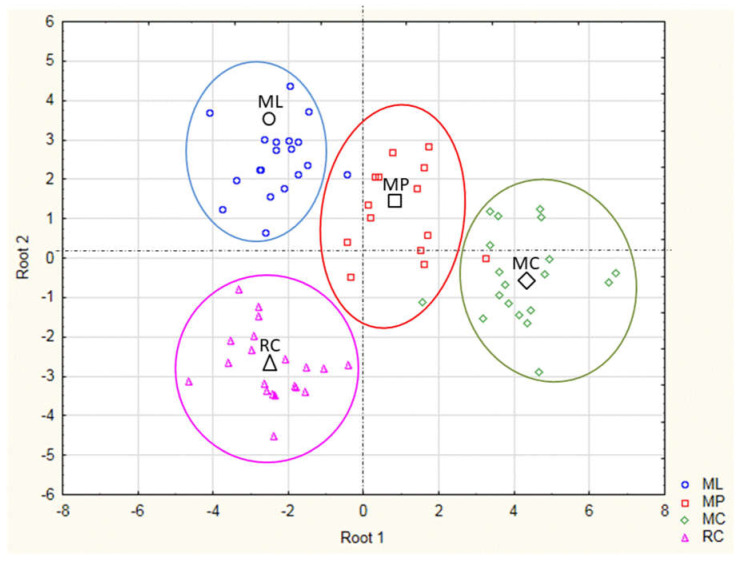
Plot of canonical discriminant analysis between ML (Mallorquina suckling lambs), MP (Mallorquina light lambs on pasture), MC (Mallorquina light lambs on concentrate) and RC (Red Mallorquina light lambs on concentrate) according to different production systems on the Balearic Islands.

**Table 1 foods-11-01312-t001:** Main supplementary feeds during fattening period and their chemical composition in Mallorquina and Roja Mallorquina lambs.

	Chemical Composition (%, DM Basis)				
	DM	Raw Protein	Raw Fibre	Ether Extract	Ash
Oat grain	88.95	10.40	12.13	3.92	2.86
Wheat grain	88.64	9.38	2.00	2.00	1.35
Concentrate supplement ^1^	87.78	17.00	7.35	3.82	2.51

^1^ Ingredients (%): barley (35.0), corn (19.0), wheat bran flour (18.8), wheat germ (10.0), sunflower (9.74), calcium carbonate (1.85), palm oil (1.50), cane molasses (1.50), salt (0.7), urea (0.7), sodium bicarbonate (0.5), mineral and vitamin correctors (0.5), magnesium oxide (0.1), bicalcium phosphate (0.02).

**Table 2 foods-11-01312-t002:** Descriptive statistics (mean, maximum and minimum) for meat-quality characteristics in Mallorquina lambs suckling (ML), grazing (MP) or supplemented with grain (MC); and Roja Mallorquina lambs supplemented with grain (RC).

	ML (*n* = 20)			MP (*n* = 20)			MC (*n* = 20)			RC (*n* = 20)			SEM	*p*-Values
	Mean	Max.	Min.	Mean	Max.	Min.	Mean	Max.	Min.	Mean	Max.	Min.		
Live weight (kg)	12.9	15.0	12.0	13.8	16.0	10.5	20.00	20.7	16.2	22.2	24.3	17.3	0.088	≤0.001
Carcass weight (kg)	6.71 ^c^	8.43	5.35	7.15 ^b^	8.11	5.82	9.31 ^a^	11.29	7.23	10.1 ^a^	13.36	6.97	0.23	≤0.001
pH_24hours_	5.89 ^b^	6.40	5.44	5.64 ^b^	6.22	5.00	5.89 ^b^	6.57	5.22	6.08 ^a^	6.59	5.78	0.04	≤0.001
*Chemical composition (% fresh meat)*														
Moisture	75.62	76.79	74.25	75.59	77.16	74.18	75.96	77.07	74.88	76.15	77.15	74.94	0.08	0.050
Protein	20.88 ^a^	22.22	19.89	20.82 ^a^	22.36	18.85	20.62 ^ab^	21.87	19.60	20.27 ^b^	21.04	19.51	0.09	0.046
Fat	1.32	2.25	0.73	1.52	3.09	0.48	1.34	2.05	0.77	1.28	2.03	0.74	0.05	0.480
Ash	1.32	1.67	1.15	1.54	2.36	1.20	1.48	2.01	1.03	1.48	2.56	1.15	0.03	0.166
*Colour*														
Myoglobin content (mg/g fresh meat)	3.68	6.78	1.46	4.02	7.08	2.04	4.06	7.29	1.82	3.04	5.00	0.86	0.16	0.084
Lightness (L*)	39.58	43.11	36.55	39.16	42.65	36.42	38.75	43.24	35.63	38.40	43.79	34.66	0.23	0.289
Redness (a*)	11.25	12.65	8.65	11.18	13.36	8.22	11.44	14.24	8.59	11.41	12.93	8.34	0.15	0.928
Yellowness (b*)	6.29	7.98	4.98	6.09	7.27	4.59	6.02	7.34	4.90	5.68	7.97	4.43	0.09	0.108
Chroma (C*)	12.92	14.33	9.98	12.77	14.87	10.78	12.98	15.26	10.98	12.80	13.94	10.30	0.13	0.937
Hue angle (H°)	29.26	38.81	22.98	28.87	41.49	21.49	28.12	40.51	21.08	26.61	37.49	19.11	0.56	0.334
Shear force (kg/cm^2^)	4.94 ^b^	7.64	2.07	5.67 ^a^	7.53	2.27	3.78 ^b^	5.30	7.27	2.87 ^c^	5.35	7.42	2.40	0.042
*Water holding capacity*														
Cooking losses	24.38	30.17	14.28	24.92	32.31	15.25	24.98	29.08	18.75	23.23	31.07	13.84	0.45	0.473
Pressure losses	13.21 ^b^	15.97	9.86	13.53 ^b^	18.37	9.49	16.38 ^a^	19.70	10.03	15.73 ^a^	19.88	12.10	0.30	≤0.001

Different superscripts (^a, b, c^) indicate significant differences (*p* < 0.05) among treatments. SEM: standard error of the mean.

**Table 3 foods-11-01312-t003:** Fatty acids identified (mean, maximum and minimum values expressed as % of the total fatty acid detected of intramuscular fat) from Mallorquina lambs suckling (ML), grazing (MP) and supplemented with grain (MC); and Roja Mallorquina lambs supplemented with grain (RC).

	ML (*n* = 20)			MP (*n* = 20)			MC (*n* = 20)			RC (*n* = 20)			SEM	*p*-Values
	Mean	Max.	Min.	Mean	Max.	Min.	Mean	Max.	Min.	Mean	Max.	Min.		
SFA	45.65 ^b^	52.54	40.82	49.38 ^ab^	55.49	42.08	50.43 ^a^	53.71	46.29	46.85 ^b^	49.96	43.58	0.450	≤0.001
C8:0-C13:0	1.71 ^c^	2.98	0.75	2.66 ^a^	3.76	1.38	2.35 ^ab^	3.49	1.49	2.02 ^bc^	3.81	0.91	0.094	0.002
C14:0	3.98 ^b^	5.64	2.34	4.82 ^a^	5.99	3.74	3.46 ^b^	4.78	2.57	3.95 ^b^	5.61	2.65	0.114	≤0.001
C16:0	21.82 ^c^	25.57	18.15	24.88 ^ab^	28.90	21.48	26.09 ^a^	27.85	24.17	22.70 ^bc^	24.91	20.91	0.303	≤0.001
C18:0	15.96 ^a^	19.13	13.89	14.70 ^b^	16.12	12.10	16.33 ^a^	19.32	13.81	16.09 ^a^	19.73	13.65	0.177	0.010
Others SFA	2.18	3.51	1.45	2.14	2.74	1.47	2.19	2.87	1.691	2.09	3.05	1.73	0.044	0.851
MUFA	34.61 ^a^	37.61	29.52	29.90 ^b^	36.98	22.93	33.53 ^a^	39.17	28.5	34.64 ^a^	39.64	30.34	0.412	≤0.001
C16:1	1.70	2.05	1.13	1.78	2.64	1.31	1.69	2.09	1.38	1.68	2.10	1.26	0.018	0.864
Total C18:1	30.13 ^a^	32.39	24.13	25.08 ^b^	31.44	18.80	28.83 ^a^	33.55	24.84	30.44 ^a^	35.22	26.67	0.404	≤0.001
Others MUFA	1.647 ^b^	2.131	1.365	1.90 ^a^	2.78	1.5	1.61 ^b^	2.47	1.132	1.74 ^ab^	1.98	1.42	0.036	0.024
PUFA	19.74 ^a^	23.54	15.08	19.59 ^a^	23.65	15.65	16.04 ^c^	19.49	13.14	17.87 ^b^	20.05	15.08	0.335	≤0.001
Total C18:2 n-6	8.89 ^b^	11.82	6.64	9.91 ^a^	12.28	8.29	8.86 ^b^	10.39	6.947	8.56 ^b^	10.60	6.36	0.160	0.027
C18:3 n-3	1.61 ^a^	2.92	0.62	1.28 ^a^	2.59	0.58	0.50 ^b^	1.00	0.254	1.41 ^a^	2.58	1.12	0.081	≤0.001
C20:2	0.19 ^b^	0.34	0.13	0.24 ^a^	0.32	0.21	0.18 ^b^	0.26	0.115	0.20 ^ab^	0.34	0.08	0.003	0.030
C20:3 n-6	0.13 ^b^	0.20	0.07	0.16 ^b^	0.26	0.06	0.18 ^ab^	0.61	0.083	0.24 ^a^	0.57	0.06	0.012	0.011
C20:4 n-6	4.25	5.96	2.58	4.31	5.85	2.94	4.18	5.90	2.943	4.15	5.48	2.59	0.098	0.948
C20:5 n-3	1.20 ^a^	2.13	0.46	0.87 ^a^	1.72	0.43	0.37 ^b^	0.54	0.107	0.97 ^a^	2.07	0.51	0.057	≤0.001
C22:5 n-3	1.50 ^a^	2.20	0.66	1.23 ^b^	1.77	0.70	0.75 ^c^	0.98	0.418	1.45 ^a^	2.20	1.02	0.048	≤0.001
C22:6 n-3	0.97 ^a^	1.68	0.57	0.55 ^b^	0.84	0.39	0.32 ^c^	0.47	0.094	0.64 ^b^	1.01	0.32	0.066	≤0.001
Others PUFA	2.73 ^ab^	4.73	1.70	2.98 ^a^	3.98	2.19	2.46 ^ab^	3.66	1.429	2.23 ^b^	3.20	1.63	0.093	0.022
n-6/n-3	2.71 ^b^	5.28	1.45	4.16 ^b^	5.85	1.91	7.11 ^a^	13.00	4.189	2.94 ^b^	3.76	1.71	0.280	≤0.001
AI	0.73 ^c^	0.94	0.50	0.93 ^a^	1.08	0.72	0.83 ^b^	1.01	0.694	0.74 ^c^	0.88	0.50	0.027	≤0.001

Different superscripts (^a, b, c^) indicate significant differences (*p* < 0.05) among treatments; SFA: saturated fatty acids; MUFA: monounsaturated fatty acids; PUFA: polyunsaturated fatty acids; AI: atherogenicity index [C12:0 + 4 × 14:0 + C16:0]/[MUFA + PUFA]; SEM: standard error of the mean.

**Table 4 foods-11-01312-t004:** Volatile compounds species identified (expressed in %) in intramuscular fat from Mallorquina lambs suckling (ML), grazing (MP), supplemented with grain (MC); and Roja Mallorquina lambs supplemented with grain (RC).

	EM	RI	LRI	ML (*n* = 20)	MP (*n* = 20)	MC (*n* = 20)	RC (*n* = 20)	SEM	*p*-Values
** *Aliphatic aldehydes* **									
3-Methylbutanal	+	+	923	7.30 ^ab^	9.43 ^b^	4.20 ^b^	17.81 ^a^	0.256	0.005
Pentanal	+	+	985	2.04	2.47	2.42	2.84	0.135	0.205
Hexanal	+	+	1.088	46.82 ^ab^	53.44 ^a^	60.98 ^a^	37.01 ^b^	2.625	0.006
Heptanal	+	+	1.193	3.97	3.76	3.42	3.40	0.174	0.560
Octanal	+	+	1.297	1.80 ^a^	1.37 ^b^	1.60 ^ab^	1.26 ^b^	0.078	0.053
Nonanal	+	+	1.403	6.31 ^a^	4.04 ^b^	4.34 ^b^	3.99 ^b^	0.008	≤0.001
Octenal	+	+	1441	0.09 ^a^	0.08 ^a^	0.07 ^a^	0.04 ^b^	0.005	0.004
Decanal	+	+	1508	0.11 ^b^	0.10 ^b^	0.08 ^b^	0.20 ^a^	0.015	0.025
(Z)-2-nonenal	+	+	1549	0.14 ^a^	0.14 ^a^	0.08 ^b^	0.16 ^a^	0.010	0.025
2(E),4(Z)-undecadienal	+		1781	0.22	0.17	0.14	0.25	0.022	0.314
2(E),4(Z)-dodecadienal	+		1824	0.03	0.04	0.04	0.03	0.047	0.333
** *Aliphatic ketones* **									
2-Butanone	+	+	906	0.57 ^b^	0.45 ^b^	0.46 ^b^	1.18 ^a^	0.101	0.021
2,3-pentanedione	+		704	1.16	0.86	1.0	1.79	0.109	0.136
2-Heptanone	+	+	1.190	0.35	0.59	0.43	0.48	0.071	0.701
2-Octanone	+	+	1.292	0.07 ^ab^	0.09 ^ab^	0.03 ^b^	0.11 ^a^	0.147	0.020
3-Hydroxy-2-butanone	+	+	1.301	2.05 ^a^	0.40 ^c^	1.24 ^b^	1.30 ^b^	0.179	0.021
1-Octen-3-one	+	+	1310	0.15	0.13	0.12	0.10	0.008	0.146
2-Hydroxy-2-propanone	+	+	1.320	0.21 ^ab^	0.17 ^ab^	0.11 ^b^	0.27 ^a^	0.022	0.057
2,3-Octanedione	+	+	1.333	2.75 ^a^	2.43 ^a^	2.57 ^a^	0.72 ^b^	0.071	≤0.001
2-Nonanone	+	+	1.397	0.07 ^ab^	0.09 ^ab^	0.05 ^b^	0.30 ^a^	0.229	0.040
2-Propanone	+		1.479	0.09	0.08	0.06	0.10	0.010	0.505
2-Decanone	+	+	1.504	0.03 ^b^	0.05 ^ab^	0.03 ^b^	0.11 ^a^	0.258	0.025
2-Undecanone	+	+	1.626	0.01	0.02	0.01	0.01	0.002	0.332
Butyrolactone	+	+	1.657	0.93	0.49	0.59	1.08	0.111	0.198
** *Aromatic hydrocarbons* **									
Toluene	+	+	1.046	0.11 ^b^	0.11 ^b^	0.08 ^b^	0.22 ^a^	0.015	≤0.001
P-Xylene	+	+	1.149	0.13	0.17	0.15	0.29	0.017	0.003
Benzaldehyde	+	+	1.544	1.24 ^ab^	1.02 ^b^	0.72 ^b^	1.80 ^a^	0.109	0.002
Benzeneacetaldehyde	+	+	1.666	0.14	0.11	0.09	0.15	0.013	0.295
1,3-Benzenediol, 4-ethyl	+		1.695	0.08	0.06	0.05	0.06	0.005	0.272
** *Aliphatic alcohols* **									
1-Penten-3-ol	+	+	1.167	0.22 ^a^	0.16 ^b^	0.07^c^	0.09 ^bc^	0.108	≤0.001
1-Pentanol	+	+	1.258	1.13	1.05	1.23	1.13	0.070	0.859
1-Hexanol	+	+	1.361	2.02	0.78	0.83	0.95	0.072	0.800
2-Ethyl-1-hexanol	+	+	1.498	0.37 ^b^	0.30 ^b^	0.27 ^b^	1.30 ^a^	0.136	≤0.001
1-Octen-3-ol	+	+	1.458	2.03 ^a^	1.57 ^a^	1.62 ^a^	1.06 ^b^	0.031	≤0.001
Heptanol	+		1.464	0.26	0.17	0.20	0.18	0.055	0.140
1-Octanol	+	+	1.567	0.35 ^a^	0.22 ^b^	0.22 ^b^	0.26 ^ab^	0.018	0.025
2-Octen-3-ol	+	+	1458	0.13 ^a^	0.12 ^a^	0.12 ^a^	0.07 ^b^	0.007	0.015
2,3-Butanediol	+	+	1592	0.78 ^a^	0.06 ^b^	0.74 ^a^	0.17 ^a^	0.244	0.044
** *Furans* **									
2-Pentylfuran	+	+	1.239	0.57 ^a^	0.71 ^a^	0.40 ^ab^	0.35 ^b^	0.079	0.019
Furfural	+	+	1.482	0.03 ^b^	0.03 ^ab^	0.03 ^b^	0.06 ^a^	0.152	0.030
2-Acethylfuran	+	+	1.524	0.04	0.05	0.04	0.08	0.008	0.242
2-Furanmethanol	+	+	1.679	0.05	0.08	0.07	0.11	0.154	0.190
** *Sulphur compounds* **									
Dimethyl disulphyde	+	+	1.081	0.72 ^b^	0.20 ^b^	0.21 ^b^	1.31 ^a^	0.211	0.351
1-Propene-1-thiol	+		1.201	3.54 ^a^	4.02 ^a^	1.96 ^b^	3.95 ^a^	0.283	0.032
Dimethyl sulfone	+	+	1.914	0.15 ^b^	0.13 ^b^	0.06 ^b^	0.26 ^a^	0.160	≤0.001
** *Pyrazines* **									
Methylpyrazine	+	+	1.276	0.40 ^b^	0.51 ^ab^	0.29 ^b^	0.84 ^a^	0.074	0.038
2,5-Dimethylpyrazine	+	+	1.332	1.45	1.44	1.26	2.37	0.234	0.307
2,6-Dimethylpyrazine	+	+	1.338	0.42	0.56	0.46	0.87	0.095	0.308
2,3-Dimethylpyrazine	+	+	1.357	0.18	0.18	0.13	0.31	0.030	0.170
2-Ethyl-6-methylpyrazine	+	+	1.395	0.21 ^a^	0.31 ^a^	0.08 ^b^	0.29 ^a^	0.186	0.022
Trimethylpyrazine	+	+	1.413	1.24	1.18	1.13	1.90	0.151	0.120
2-Ethyl-3-methylpyrazine	+	+	1.418	0.32	0.29	0.15	0.21	0.085	0.207
3-Ethyl-2,5-dimethylpyrazine	+	+	1.454	0. 50 ^b^	0.59 ^ab^	0.38 ^b^	1.04 ^a^	0.086	0.025
2-Ethil-3,5-dimethylpyrazine	+	+	1.471	0.18	0.22	0.16	0.33	0.029	0.148
2-Methyl-3,5-diethylpyrazine	+	+	1.504	0.08	0.09	0.07	0.13	0.018	0.706
** *Pyrroles* **									
Pyrrole	+		1.536	0.10 ^b^	0.14 ^b^	0.12 ^b^	0.26 ^a^	0.024	0.004
3-Methyl-pyrrole	+		1.592	0.29	0.01	traces	0.15	0.066	0.141
2-Acethylpyrrole	+	+	1.967	0.22	0.18	0.18	0.35	0.027	0.061
** *Carboxylic acids* **									
Acetic acid	+	+	1.486	0.90	0.55	0.68	1.02	0.098	0.336
2-Hydroxy-propanoic acid	+		1.556	0.07	0.05	0.04	0.08	0.010	0.636
Propionic acid	+	+	1.570	0.06	0.05	0.04	0.13	0.024	0.567
Hexanoic acid	+	+	1.862	0.21	0.17	0.15	0.20	0.017	0.543
2-Ethylhexanoic acid	+	+	1.944	0.01	0.01	traces	0.01	0.003	0.900
Heptanoic acid	+	+	1.949	0.02 ^b^	0.02 ^b^	0.02 ^b^	0.05 ^a^	0.167	0.033
Octanoic acid	+	+	2.031	0.05 ^b^	0.03 ^b^	0.06 ^b^	0.10 ^a^	0.123	0.017
Decanoic acid	+		2.105	0.13	0.14	0.12	0.25	0.135	0.065

Different superscripts (^a, b, c^) indicate significant differences (*p* < 0.05) among treatments; EM: tentatively identified by NIST/EPA/NIH libraries; RI: approximate identification by comparing the retention index with literature values to TR polar DB-WAX column; LRI: linear retention indices calculated similarly to Kovats [49] ones.

**Table 5 foods-11-01312-t005:** Evaluations of sensorial analysis by trained panellists, for meat quality characteristics in Mallorquina lambs suckling (ML), grazing (MP), supplemented with grain (MC); and Roja Mallorquina lambs supplemented with grain (RC).

	ML (*n* = 20)	MP (*n* = 20)	MC (*n* = 20)	RC (*n* = 20)	SEM	*p*-Values
Lamb odour	5.96	5.99	6.19	6.04	0.099	0.852
Milk odour	6.97	6.34	6.52	6.76	0.125	0.323
Lactic acid flavour	5.89 ^a^	4.95 ^b^	5.58 ^ab^	5.95 ^a^	0.130	0.030
Lamb flavour	6.03 ^b^	6.94 ^a^	6.28 ^b^	6.38 ^b^	0.099	0.014
Liver flavour	1.29 ^b^	2.10 ^a^	2.05 ^ab^	1.28 ^b^	0.138	0.041
Global flavour persistence	4.86 ^c^	6.04 ^a^	4.66 ^c^	5.05 ^b^	0.123	≤0.001
Hardness	3.64	3.44	3.78	3.68	0.096	0.682
Initial juiciness	5.94	6.31	5.98	6.22	0.072	0.218
Final juiciness	6.75 ^b^	7.31 ^a^	7.10 ^a^	7.23 ^a^	0.078	0.049
Friability	7.39 ^b^	7.93 ^a^	7.27 ^b^	7.32 ^b^	0.087	0.035
Chewiness	7.69	8.01	7.44	7.63	0.085	0.139

Different superscripts (^a, b, c^) indicate significant differences (*p* < 0.05) among treatments; SEM: standard error of the mean.

**Table 6 foods-11-01312-t006:** Summary of stepwise regression for organoleptic and nutritional variables to discriminate Mallorquina and Red Mallorquina from a traditional production system on the Spanish Mediterranean Islands.

	*F*-Values	*p*-Values
∑ Polyunsaturated fatty acids	24.987	≤0.001
C18:2n-6t	23.379	≤0.001
C18:2n-6c	20.502	≤0.001
Conjugated linoleic acid	16.395	≤0.001
C18:1n-9c	16.133	≤0.001
C17:1	12.280	≤0.001
Carcass weight	11.715	≤0.001
Trans-vaccenic acid	9.487	≤0.001
C10:0	7.460	≤0.001
Water holding capacity	7.072	≤0.001
Aromatic hydrocarbons	6.406	0.009
C16:1	5.998	≤0.001
C18:0	5.928	≤0.001
C12:0	5.097	0.003
Final juiciness	4.846	0.005
C22:1 n-9	4.550	0.007
C23:0	4.235	0.009

## Data Availability

Data is contained within the article.

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
