# Peer review of "Organoleptic and Nutritional Traits of Lambs from Spanish Mediterranean Islands Raised under a Traditional Production System"

_foods, 2022, doi:10.3390/foods11091312_

Round 1

Reviewer 1 Report

The paper belongs to original scientific paper and presents very useful data of organoleptic and nutritional properties of meat from suckling and light lambs in local breeds on the Spanish Mediterranean islands. There are many papers on this topic, but this one gives us information about a specific locality.

Unfortunately, I didn't have time to check this paper for plagiarism.

In section: Materials and methods: Is there a newer standard compared to the one listed? For sensory evaluation should be added and International Organization for Standardization – ISO. (2006). ISO 5496: Sensory analysis: Methodology: Initiation and training of assessors in the detection and recognition of odours. Genéve: ISO?

References should be given in alphabetical order?

Reference number 2 cannot be found.

Author Response

Dear reviewer,

Thank you so much for your contribution. We have listened to your suggestions and we send the corrections to the document.

Best regards

Alberto Horcada

Question 1. Table 2. Perhaps it would be better if the authors present standard deviation as to min and max values here.

Thanks for your suggestion. The authors have presented the maximum and minimum values to have a real descriptive range of values. The soul of the document is to make a description of the values that could be observed in the studied parameters.

Question 2: p-values should just be up to 3 decimal place.

Changes have been made throughout the text.

Question 3: Was the panel trained? How did the authors arrive with these sensory attributes? More information needed. 

Changes in lines 247 to 253 has bee reported.

Question 4: No info on the multivariate canonical DA approach, please add.

Text about multivariate canonical DA approach has been included in 2.6. Statistical analysis Section.

Question 5: I'd recommend the authors to also provide justification why and the aim of the following analyses has been performed in Section 2.6 

Section 2.6 shows the statistical methods used to achieve the objective of the work, which is to characterize the meat of Mediterranean breed lambs in four production models. On the other hand, it is intended to present the variables that contribute to the greatest extent to the differentiation of the different animal groups. The methodology used to achieve the objectives is also explained.

Changes in 2.6 Section have been performed

Question 6: What are the limitations and future research avenues? Please add  

Changes in 4. Conclusion Section had been reported

Reviewer 2 Report

Table 2. Perhaps it would be better if the authors present standard deviation as to min and max values here.

p-values should just be up to 3 decimal place.

Was the panel trained? How did the authors arrive with these sensory attributes? More information needed. 

No info on the multivariate canonical DA approach, please add.

I'd recommend the authors to also provide justification why and the aim of the following analyses has been performed in Section 2.6 

What are the limitations and future research avenues? Please add    

Author Response

Dear reviewer,

Thank you so much for your contribution. We have listened to your suggestions and we send the corrections to the document.

Best regards

Alberto Horcada

Manuscript Title: Organoleptic and nutritional traits of lambs from Spanish Mediterranean islands raised under a traditional production system.
The study characterized the organoleptic and nutritional properties using meat from two autochthonous breeds in danger of extinction linked to the Spanish Mediterranean island environment. This manuscript is meaningful and interesting. However, some errors are scattered throughout the text, and a closer editorial editing is needed for it to be published. The following are a few examples.

  1. Line 16: In your abstract, what is the meaning of “suckling and light lambs”, it is necessary to add brief description to make readers clear.

The space in the abstract is limited to 200 words. A brief description of suckling and light lambs has been made. A more extensive description has been made in the Introduction section. Publisher rules probably remove the description made in the abtract.

(See line 17).

  1. Line 22-23: “We analysed pH, colour, texture, water holding capacity, fatty acids, volatile compounds and sensorial attributes of the meat”→“pH, colour, texture, water holding capacity, fatty acids, volatile compounds and sensorial attributes of the meat were analysed ”.

Change in Abstract Section had been reported (lines 23 to 25)

  1. Line 23-24: Please supplement the results about meat quality traits such as WBSF.

The authors have considered your proposal. Unfortunately, results for Colour and WBSF are not shown because space in the abstract is limited to 200 words and no significant differences between treatments have been observed.

  1. Line 102: For meat quality traits like pH and colour would be affected with slaughter age/day. Are all lambs slaughtered on the same day? So more details should be added about this.

We agree, the effect of the day could affect to pH and colour because stress of animals in the abattoir is very important on meat quality. However, all animals were not sacrificed on the same day. We controled usual restrictions in reference to the welfare animals in the abattoir. It is very complicated to slaughter 80 carcasses in the slaughterhouse in the same day. Then, sacrifices were scheduled in batches of 20 animals. The animals were sacrificed around similar age in each group. To avoid random effects Carcass weight was used as a linear covariate. This is detailed in Section 2.6. Statistical analysis.

  1. Line 109: How  20 lambs in each group selected, in other words, what is the basis for the selection?

We select based on similar weights and age. A sentence in 2. Materials and Methods Section has been included. (lines 107 and 108)

  1. Line 119: What is the range of movement of each lamb in each group?

Ranges of live weight of lambs were included in table 2.

  1. How to kill animals? Does it meet the requirements of animal welfare?

This information was previously shown at the beginning of Section 2. Reference [10] according to the guidelines of Council Regulation on the protection of animals at the time of slaughtering. (Lines 109 and 110)

  1. What is the driving distance from farm to slaughterhouse, and does stress take into account?

The distance to slaughterhouses was not controlled. The sacrifice of the animals was carried out in the local slaughterhouse of Marratxí (07141 Mallorca, Spain). The distance from the farms to the slaughterhouse did not exceed 50 km because the island of Mallorca is small. Animal stress did not occur. Transport and pre-slaughter stress did not affect meat quality because the pH values of the meat samples were normal (Table 2).

  1. Line141-144: The description of the pH measurement method is too simple. To ensure the reproducibility of the experiment, the pH measurement method has to be described in full detail.For example, how is the meter calibrated, do you apply temperature compensation?

Description of the method to measure pH has been carried out in 2.2 Section. Lines 148-150. 

  • Line 165: How many times are measured about WBSF ? Please specify.

Because samples of longissimus thoracis – lumborum is small analyses were run in duplicate. It is show in line 178.

  • Line 190-191: The “30C/min” and “20C/min” should be revised to “30℃/min” and “20℃/min”, respectively.

It is true.They were several mistakes. Changed were made in lines 197 to 199.

  • Line 294: Why the Roja Mallorquina breed is probably more susceptible to stress factors at the slaughterhouse than the Mallorquina. Please add more details.

This is an idea proposed by the authors to justify the results according to reference 25 of the text. The authors have not studied the effect of stress on the Majorcan and Red Majorcan breeds. The text says: .... leading to a late increase in the pH of the meat 25. (line 316), Probably genetic factors could be considered.

  • Line 314: There are thresholds published for meat colour and shear force which can put the practical significance of the results in context.

Reference [22] was included in line 331.

  • Line 318: “Although the myoglobin content observed in the meat of the two suckling lambs (3.68 mg/g fresh meat)”, what the meaning of the “two”?

It was a mistake. A new text lines were included in lines 338 to 341.

  • Line 325: Missing symbols between “observed”and “It”.

Symbol in line 341 was included.

  • Line 328: “Results observed to trichromatic values for meat in light lambs in the Mallorquina and Roja Mallorquina breeds are within the range reported by Mateo et al.”should be revised to “Results observed to trichromatic values for meat in light lambs in the Mallorquina and Roja Mallorquina breeds were within the range reported by Mateo et al.”

Lines 349 and 350 has been rewritten

  • Line 337: “No differences in lightness, redness, yellowness, and Chroma and hue were observed in meat among suckling lambs, grazing and concentrate feeding lamb groups”, the “and”between “yellowness” and “Chroma” should be removed, is the change right?

Line 358 the change was made. We agree.

  • Line 346: “The shear force values of the meat from all the feeding systems treatment agree withthose reported by Mateo et al” should be revised to “The shear force values of the meat from all the feeding systems treatment agreed with those reported by Mateo et al”.

Lines 368 and 369 the change was made. We agree

  • Line 366: “which agrees with the lack of differences in cooking water losses observed”should be revised to “which agreed with the lack of differences in cooking water losses observed”, please pay attention to similar error.

Gramatical mistake has been revised. Line 388 has been changed.

  • Line 446: The threshold published for each flavor substance is different, which contribute to the degree of contribution to the meat is also different. Therefore, it is necessary to use the ROAV to screen the key flavor substances.

Excuse me, but we don't understand the line 446 reference. In line 446 the authors are talking about  fat nutritional indexes (AI index). If you are referring to the use of relative odor activity value (ROAV), the authors have not performed an analysis of Volatiles compounds olfactory activity (in 3.5. Section). In fact, the authors have based themselves on thresholds reported by the bibliography consulted.

  • Line 466: Aliphatic alcohols or aliphatic alcohols?

Change in line 489 has been done (aliphatic alcohols).

  • Line 474: “The relative content of octanal, nonanal and 3-hydroxy-2-butanone has been high in the meat of all the animals studied”should be revised to “The relative content of octanal, nonanal and 3-hydroxy-2-butanone have been high in the meat of all the animals studied”, is the change right?

We think "has been" is correct because it is singular. The third singular person ('The relative content')

  • Line 646: There are a number of formatting errors in the references, please check and correct each one.

References Section has been revised and formal mistakes has been rewritten.

Reviewer 3 Report

Comments to the Author

Manuscript Title: Organoleptic and nutritional traits of lambs from Spanish Mediterranean islands raised under a traditional production system
The study characterized the organoleptic and nutritional properties using meat from two autochthonous breeds in danger of extinction linked to the Spanish Mediterranean island environment. This manuscript is meaningful and interesting. However, some errors are scattered throughout the text, and a closer editorial editing is needed for it to be published. The following are a few examples.

  1. Line 16: In your abstract, what is the meaning of “sucklingand light lambs”, it is necessary to add brief description to make readers clear.
  2. Line 22-23: “We analysed pH, colour, texture, water holding capacity, fatty acids, volatile compounds and sensorial attributes of the meat”→“pH, colour, texture, water holding capacity, fatty acids, volatile compounds and sensorial attributes of the meat were analysed ”.
  3. Line 23-24: Please supplement the results about meat quality traits such as WBSF.
  4. Line 102: For meat quality traits like pH and colour would be affected with slaughter age/day. Are all lambs slaughtered on the same day? So more details should be added about this.
  5. Line 109: How arethe 20 lambs in each group selected, in other words, what is the basis for the selection?
  6. Line 119: What is the range of movement of each lamb in each group?
  7. How to kill animals? Does it meet the requirements of animal welfare?
  8. What is the driving distance from farm to slaughterhouse, and does stress take into account?
  9. Line141-144: The description of the pH measurement method is too simple. To ensure the reproducibility of the experiment, the pH measurement method has to be described in full detail.For example, how is the meter calibrated, do you apply temperature compensation?
  10. Line 165: How many times are measured about WBSF ? Please specify.
  11. Line 190-191: The “30C/min” and “20C/min” should be revised to “30℃/min” and “20℃/min”, respectively.
  12. Line 294: Why the Roja Mallorquina breed is probably more susceptible to stress factors at the slaughterhouse than the Mallorquina. Please add more details.
  13. Line 314: There are thresholds published for meat colour and shear force which can put the practical significance of the results in context.
  14. Line 318: “Although the myoglobin content observed in the meat of the two suckling lambs (3.68 mg/g fresh meat)”, what the meaning of the “two”?
  15. Line 325: Missing symbols between “observed”and “It”.
  16. Line 328: “Results observed to trichromatic values for meat in light lambs in the Mallorquina and Roja Mallorquina breeds are within the range reported by Mateo et al.”should be revised to “Results observed to trichromatic values for meat in light lambs in the Mallorquina and Roja Mallorquina breeds were within the range reported by Mateo et al.”
  17. Line 337: “No differences in lightness, redness, yellowness, and Chroma and hue were observed in meat among suckling lambs, grazing and concentrate feeding lamb groups”, the “and”between “yellowness” and “Chroma” should be removed, is the change right?
  18. Line 346: “The shear force values of the meat from all the feeding systems treatment agree withthose reported by Mateo et al” should be revised to “The shear force values of the meat from all the feeding systems treatment agreed with those reported by Mateo et al”.
  19. Line 366: “which agrees with the lack of differences in cooking water losses observed”should be revised to “which agreed with the lack of differences in cooking water losses observed”, please pay attention to similar error.
  20. Line 446: The threshold published for each flavor substance is different, which contribute to the degree of contribution to the meat is also different. Therefore, it is necessary to use the ROAV to screen the key flavor substances.
  21. Line 466: Aliphatic alcohols or aliphatic alcohols?
  22. Line 474: “The relative content of octanal, nonanal and 3-hydroxy-2-butanone has been high in the meat of all the animals studied”should be revised to “The relative content of octanal, nonanal and 3-hydroxy-2-butanone have been high in the meat of all the animals studied”, is the change right?
  23. Line 646: There are a number of formatting errors in the references, please check and correct each one.

Author Response

Dear reviewer,

Thank you so much for your contribution. We have listened to your suggestions and we send the corrections to the document.

Best regards

Alberto Horcada

I read this paper with great interest. I would congratulate the authors for the efforts and quality of this study that is more than important scientifically and objectively to achieve the sustainable developments objectives.

Thank you very much. We intend to revalue animal production and conserve autochthonous sheep breeds.

The methods are well described, and the discussion is weak. I would recommend to the authors to further detail the discussion and improve its design. I think there is an opportunity to organize further the discussion. in terms of future outcomes, the conclusion needs improvements and the drawbacks need also to be highlighted. As an open perspective, I would suggest for the authors to mention the potential of omics for a better characterization of these autochthonous breeds. For instance, proteomics can be one omic approach that can be used, refer to: https://doi.org/10.1016/j.meatsci.2020.108311

https://doi.org/10.1016/j.jprot.2012.04.011

https://doi.org/10.1016/j.jprot.2012.10.023

We agree. Thank you so much for your suggestion. The authors report that this is the first time that these two local breeds have been worked on to characterize the quality of the meat and the production system in the Balearic Islands (Spain). Therefore, the first characterization results of the two breeds are presented. Currently the number of sheep of the two breeds is small. To guarantee the increase of animals of the Majorcan and Red Majorcan breeds, improvement plans are being developed. The authors are going to propose proteomic studies to get to know the quality of the meat and characterize the traditional production systems of the Balearic Islands.

Lines 626 to 639 were included in the text (3. Results and discussion Section)

Lines 658 to 664 were included in the text (4. Conclusion Section)

Reviewer 4 Report

I read this paper with great interest. I would congratulate the authors for the efforts and quality of this study that is more than important scientifically and objectively to achieve the sustainable developments objectives.

The methods are well described, and the discussion is weak. I would recommend to the authors to further detail the discussion and improve its design. I think there is an opportunity to organize further the discussion. in terms of future outcomes, the conclusion needs improvements and the drawbacks need also to be highlighted. As an open perspective, I would suggest for the authors to mention the potential of omics for a better characterization of these autochthonous breeds. For instance, proteomics can be one omic approach that can be used, refer to: https://doi.org/10.1016/j.meatsci.2020.108311

https://doi.org/10.1016/j.jprot.2012.04.011

https://doi.org/10.1016/j.jprot.2012.10.023

Author Response

-

Round 2

Reviewer 3 Report

The author has revised the manuscript seriously, I think the manuscript is acceptable.

Author Response

Thank you for your suggestions and revised version submitted. I think most of your suggestions had been responded.

Please consider our new minor contributions:

1. In the tables ≤0.000 should be replaced by ≤0.001

Changes had been reported.

  1. Sometimes names of compounds are in lower case or upper case, please standardize. Please, check for this within the text, for example L 655 hexanal (lower case maybe more appropriated in the text)

Changes had been reported.

  1. L27 aldehyde or aldehydes?

Changes had been reported.

  1. L76-77. If animals are grazing for 2 to 3 months after weaning, their age will reach up to 4.5 months old, which will not coincide with the expression in the abstract. Better expressed as 'with their mothers until reaching three months old...' if the sentence in the abstratc is correct. Otherwise, the abstract should be corrected.

Light lambs correspond to approximately three-four months of slaughter age as detailed in material and methods. The light lambs of the Red Mallorquina breed are traditionally slaughtered with approximately 1 month more life than those of the Mallorquina breed (detailed in material and methods). The authors have included the word ‘around’ in the abstract to describe light lambs.  Line 2 of the abstract.

  1. L87, 656. Mediterranean islands' breeds

These sentences had been revised.

  1. L162-163. -1 and 1/2 should be superscripts.

Changes had been reported.

  1. L315 In the discussion (an also should be clear in material and methods) it should be considered the fact that the animals were not slaughtered in the same day, and that potentially could have an effect in the results.

Reference to influence of the animals were not slaughtered in the same day on the results had been reported in 3.1 Section.

  1. L658-659 the sentence is repeated

Changes had been reported. 
